# OpenReview forum: "Align Once, Benefit Multilingually: Enforcing Multilingual Consistency for LLM Safety Alignment"
_ICLR.cc/2026/Conference — ICLR 2026 Poster_

### Official Review · Reviewer_4TKF · 2025-10-29

**Soundness:** 3
**Presentation:** 2
**Contribution:** 3
**Rating:** 4
**Confidence:** 3

**Summary:**

The paper proposes a plug-and-play Multilingual Consistency (MLC) loss to make safety alignment transfer across languages without collecting response-level labels in every language. During alignment, the trainer feeds translated versions of the same prompt through the model, builds a small matrix of their internal representations and adds an auxiliary loss that pushes those representations to be rank-1 and that pushes their hidden representations toward dominant singular direction. The intuition is if the model interprets what safe looks like similarly in all languages at the hidden-state level, then a single-language alignment, usually English will lift safety everywhere. The loss is implemented via a temperature-softmax over the singular values of the representation matrix, encouraging the top singular value to dominate. The goal is to close the gap in safety where multilingual safety often lags far behind English.

**Strengths:**

- The rank-1 and softmax-over-singular-values objective is a clean way to apply make language variants point the same way and can be easily integrated into standard pipelines. It is differentiable, easy to bolt on, and doesn't interfere with the main alignment loss.

- No need for response-level labels in low resource languages, prompt translations are sufficient.

- The method generalizes to unseen languages in MultiJail, supporting the claim that it regularizes a language-agnostic safety direction rather than overfitting to particular languages.

**Weaknesses:**

- The paper does not clearly surface which layer the representations are extracted from or how the extractor is designed. Also, the main text does not show a quick sensitivity table or plot to convey how these choices affect results.

- The connections between the spectral objective, its softmax relaxation over singular values and the final training loss are not well traced in the main part and that makes it hard for readers to follow the derivation without repeatedly jumping to the appendix.

- The utility evaluation is summarized briefly and the paper does not give practical guidance on how to tune the auxiliary-loss weight or temperature to manage trade-offs between safety gains and general multilingual capability.

**Questions:**

- Can you give reference to the MMMLU-lite dataset? What is that dataset?

- How should practitioners tune the auxiliary-loss weight and temperature to balance safety gains against utility?

- What happens when translations are imperfect such as noisy or partially wrong? Can you weight pairs by MT quality or be robust to mismatches?

- Does the layer choice interact with backbone architecture?

---

> ### Author Response · Authors · 2025-11-20
> **Response to Reviewer 4TKF (Part I)**
>
> We sincerely thank the reviewer for the valuable time and constructive suggestions, which have helped us improve the clarity and completeness of our paper. Below, we provide point-by-point responses and clarify the improvements made in the revised version.
>
> 1. **Layer Selection Discussion (Weakness 1 & Question 4)**: Thank you for your kind reminder. We already included the discussion in Appendix E.1 in the original submission. Following your suggestion, we surface this analysis in the main text (Section 4.7) in revision. In this part, we clarify that in our main experiments, we default to using the final hidden layer for multilingual representation extraction.  The layer choice does interact with backbone architecture. For instance, we use Layer 28 for Qwen 2.5-7B and Layer 42 for Gemma 2-9B. However, last layer is not necessarily the optimal choice. Thus we provide a dedicated sensitivity analysis in this part (Figure. 4). Using Qwen-2.5-7B-Instruct as an example, we plot how layer choice affects the trade-off between safety gains and multilingual utility. The results reveal a practical trade-off: aligning at deeper layers yields stronger safety alignment but may impair multilingual utility, while middle layers offer a better balance.
>
> 2. **Clarify the Derivation (Weakness 2)**:  We appreciate the helpful suggestion! We have revised Section 3.1 to significantly improve the reading flow by adding a clear transition between the theoretical proposition and the resulting loss formulation. Specifically, we transform the alignment objective to the spectral rank-1 objective, _i.e._, maximizing the largest singular value. This newly derived objective is then implemented via a softmax-based loss function, which constitutes our final training loss. This progressive and coherent derivation ensures that our learning objective is transparent and easy to follow.
>
> 2. **About MMMLU-lite Dataset (Question 1)**:  We have provided a detailed introduction to the MMMLU-lite dataset in Appendix D.2. MMMLU-lite is a lite version of the MMMLU dataset developed by the OpenCompass community (https://huggingface.co/datasets/opencompass/mmmlu_lite). The original MMMLU dataset contains over 200,000 multiple-choice questions translated into 14 languages.
> To reduce the dataset size for practical evaluation, MMMLU-lite uniformly samples 25 examples per subject-language pair using a fixed random seed to ensure reproducibility, resulting in a total of 19,950 examples, which is approximately 10\% of the full dataset. MMMLU-lite provides a compact yet comprehensive benchmark for evaluating the multilingual general knowledge and reasoning capabilities of large language models. We confirmed that MMMLU-lite provides a representative and reliable assessment of multilingual general knowledge and reasoning capabilities, correlating closely with the results of the full benchmark.
>
>
> 3. **About Hyperparameter Analysis (Weakness 3 & Question 2)**: We recognize the need for practical guidance on managing the trade-off between safety and utility. Thus, in Appendix E.3, we provide the analysis of hyperparameter $\lambda_{aux}$. In addition, we add the following analysis on the impact of $\tau$ (which regulates the "hardness" of the consistency constraint) for reference:
>
>     | $\tau$              |       Avg Safety Rate ↑ |        PAG ↑ | MMLU ↑ | MMMLU-lite ↑ |
>     | -------------- |  --------: | ---------: | -----: | -----------: |
>     | 0.01           |    65.28 |          0.54 |  76.27 |        54.89 |
>     | 0.1            |    91.17 |      0.8517 |  75.97 |        57.56 |
>     | 0.2  |  95.94 |       0.9697 |  76.30 |        52.66 |
>     | 0.5            |      98.28 | 0.9838 |  76.52 |    39.11 |
>
> Related experimental results will be included in revision.

---

> > ### Author Response · Authors · 2025-11-20
> > **Response to Reviewer 4TKF (Part II)**
> >
> > 4. **About Impact of Translation Quality (Question 3)**:
> > This is a very intesting question!
> > To study the effect of imperfect translations, we constructed three variants of the multilingual training data using translators of different quality levels (ranked by COMET scores):
> >
> >     * GPT-4o:  neural machine translation ranked as high-quality (used as our primary setup).
> >
> >     * NLLB: open-source machine translation ranked as mid-quality.
> >
> >     * Permute-30: a noise-augmented version of GPT-4o translations where 30% of sentences were perturbed by random truncation, insertion of irrelevant characters, or insertion of unrelated segments
> >
> >     We then trained MLC separately on each version of the translated dataset. The results are:
> >
> >     | Model      | Avg ↑     | PAG ↑     | MMLU ↑  | MMMLU-lite ↑|
> >     | ---------- | ---------  | --------- | ----- | ---------- |
> >     | GPT-4o     | 95.94     | 0.970     | 76.3  | 52.66      |
> >     | NLLB       | 96.83     | 0.973 | 76.26 | 47.81      |
> >     | Permute-30 | 96.56 | 0.961     | 76.43 | 45.84      |
> >
> >     The results show that our MLC method maintains an exceptionally high and stable safety rate (all $> 95.9\%$) and consistency (PAG $> 0.96$) regardless of the translation quality, demonstrating high robustness to noisy or partially imperfect training data. We attribute this to the fact that safety is a high-level semantic property, and our consistency objective is relatively insensitive to moderate amounts of MT noise. However, training data with greater translational noise significantly degrades multilingual general utility (MMMLU-lite score drops from 52.66 to 45.84). This suggests that while safety alignment is robust, the model's general multilingual representations are still sensitive to input quality.

---

> ### Author Response · Authors · 2025-11-27
> **Follow-Up: Seeking Further Feedback Before Rebuttal Deadline**
>
> Dear Reviewer 4TKF,
>
> Thank you again for your thoughtful and constructive feedback. We’ve carefully addressed your concerns with detailed responses and conducted additional experiments based on your suggestions. As the rebuttal deadline approaches, please let us know if you have any lingering or follow-up thoughts, as we would greatly appreciate the opportunity to clarify them and further improve the manuscript. We also sincerely appreciate your recognition of the clarity of our rank-1 objective, its ease of integration, and its ability to generalize to unseen languages without requiring response-level supervision. It is very encouraging to see these aspects highlighted among the strengths, and we are grateful that you found these contributions meaningful. If our revisions adequately address your concerns, we would be thankful for your consideration of this in your overall evaluation.
>
> Kind regards,
>
> Authors of Submission 9504.

---

### Official Review · Reviewer_4nM1 · 2025-11-01

**Soundness:** 3
**Presentation:** 3
**Contribution:** 3
**Rating:** 6
**Confidence:** 4

**Summary:**

The paper introduces a plug-and-play Multilingual Consistency loss to improve alignment across language representation and to transfer safety capabilities of LLMs from high-resource to low-resource languages. This auxiliary loss promotes consistency across languages by pushing the model to produce similar internal activations of queries written in different languages. The loss can also be integrated into various safety alignment paradigms, such as SFT and DPO. The results demonstrate improved safety performance across languages, notably closing the gap between high- and low-resource languages and largely preserves the general multilingual capability of the model.

**Strengths:**

1. The objective is intuitive, effective and does not rely on any anchor languages.
2. The auxiliary loss objective can be generalized and integrated to any post-training safety paradigms.
3. The approach improves substantially safety performance of low-resource languages, while retaining that of high-resource languages.

**Weaknesses:**

1. The approach is potentially sensitive to hyperparameters such as layer selection. The best layer where representation alignment is most effective also seems task sepcific.
2. Scaling behavior of the objective is not tested beyond 7B. Divergence across languages may be beneficial for even larger models, where the consistency objective may not be effective.

**Questions:**

1. Would the method be effective too for larger models, e.g., Qwen 14B / 32B?
2. Beyond safety applications, can the method be used to reduce low- and high-resource language gaps exist for other multilingual capabilities?
3. Could the method be harmful when handling culturally sensitive tasks? What about the possibility of altering cultural-specific knowledge in the model, which might be encoded in language-specific representation?

---

> ### Author Response · Authors · 2025-11-20
> **Response to Reviewer 4nM1 (Part I)**
>
> We sincerely thank the reviewer for the valuable feedback and for highlighting both the strengths and limitations of our work. Below we address the concerns in detail.
>
> 1. **Layer Selection Sensitivity (Weakness 1)**:  We agree that the choice of the representation extraction layer influences the alignment outcome. While intervention at any layer depth yields safety improvements compared to the unaligned baseline, the magnitude of safety gain and the impact on general utility vary significantly by layer. This ultimately presents a practical trade-off between safety and utility when selecting the optimal layer for intervention. The optimal intervention layer depth is indeed model- and task-specific.
>
>     In our main experiments, we default to using the final hidden layer for multilingual representation extraction. In Revision Section 4.7 (original Appendix E.1), we performed an ablation study to explore how layer choice affects the trade-off between safety and utility. The analysis is conducted on Qwen-2.5-7B-Instruct (28 layers).
>
>     From *Figure 4* in Section 4.7, we can see that as alignment is applied to deeper layers, both the average multilingual safety rate and cross-lingual consistency (PAG) increase substantially, reaching a high and stable level in the final layers. This suggests that deeper layers, being closer to the model’s output, are more responsive to consistency-based supervision. In contrast, multilingual utility (MMMLUlite) exhibits a non-monotonic pattern: it improves at middle-to-late layers but drops when using the final layer. This indicates that intermediate layers encode more language-agnostic semantic features and are thus better suited for multilingual alignment. The final layer, by comparison, appears more language-specific, and applying alignment constraints at this stage may disrupt multilingual generalization. Meanwhile, English utility (MMLU) remains stable across all layers, suggesting that anchor-language representations are robust throughout the model.
>
>     These findings reveal a practical trade-off: aligning at deeper layers yields stronger safety alignment but may impair multilingual utility, while middle layers offer a better balance. This highlights the importance of layer selection in achieving alignment objectives and motivates future work on layer-aware alignment strategies.
>
> 2. **Model Scaling Behavior (Weakness 1 & Question 1)**: Thank you for your suggestion on the scaling behavior testing. Section 4.4 of the main paper already evaluates Qwen-2.5 at 1.5B / 3B / 7B scales. Following the reviewer’s suggestion, we extend experiments to 14B and 32B models. Below are supplemental results:
>
>     |**Models**|**EN**|**ZH**|**RU**|**JA**|**AR**|**BN**|**SW**|**UR**|**PS**|**KU**|**Avg Safety Rate ↑**|**PAG ↑**|
>     | ------------------| ------------| ------------| -----------| ------------| ------------| -----------| -----------| -----------| -----------| -----------| ------------|  ------------|
>     |**Qwen 2.5-14B**|||||||||||||
>     |Raw|93.33|98.33|96.11|94.44|97.78|88.33|20.00|74.44|25.56|24.44|71.28|0.6120|
>     |DPO|94.44|91.67|93.33|87.22|93.30|82.78|27.78|72.63|25.00|22.91|69.11|0.6935|
>     |DPO+MLC|100.00|100.00|99.44|99.44|100.00|98.89|95.00|98.89|98.33|98.89|98.89|0.9812|
>     |**Qwen 2.5-32B**||||||||||||||
>     |Raw|96.70|97.80|95.00|94.40|95.60|91.10|39.40|80.00|31.10|24.40|74.55|0.6440|
>     |DPO|98.30|98.90|99.40|95.60|98.30|95.00|49.40|83.90|47.20|31.70|79.77|0.6960|
>     |DPO+MLC|100.00|99.40|98.30|100.00|100.00|99.40|89.40|97.20|96.10|98.90|97.87|0.9650|
>
>     We can see that larger models exhibit stronger multilingual safety in the raw setting, likely due to increased model capacity. The experimental results suggest that MLC consistently enhances multilingual safety and stability across model scales from 1.5B to 32B.
>
> 3. **General Application to Other Capabilities (Question 2)**:  Yes! We agree with you that MLC is not inherently tied to safety, indicating broader applications. Conceptually, the loss encourages cross-lingual representational alignment, which is a general principle applicable to other multilingual tasks. We emphasize this as a promising direction for future work.

---

> > ### Author Response · Authors · 2025-11-20
> > **Response to Reviewer 4nM1 (Part II)**
> >
> > 4. **Cultural Sensitivity and Knowledge Preservation  (Question 3)**: This is a very interesting and important question.
> > We fully agree that the possibility of altering or interfering with cultural-specific knowledge encoded in language-specific representations must be carefully considered.
> > Our current approach, which uses a simple yet effective linear projection to align the safety subspace, is inherently a general-purpose alignment mechanism. While our results on MMLU and MMMLU-lite(Section 4.2, Table 3) confirm that our method does not introduce significant negative impact on the model's existing general world knowledge, however, many culturally sensitive nuances are tightly coupled with specific linguistic representations.
> > The straightforward feature extraction and alignment may indeed inadvertently interfere with this culturally sensitive information, particularly when a single, monolithic feature space is enforced for safety regularization. We view this as a critical limitation and an exciting avenue for future work. We propose that future research should focus on finer-grained alignment strategies, involving techniques such as: decoupling language-specific or cultural-specific dimensions and applying selective alignment to preserve cultural richness.

---

### Official Review · Reviewer_94Ju · 2025-11-01

**Soundness:** 2
**Presentation:** 2
**Contribution:** 2
**Rating:** 6
**Confidence:** 2

**Summary:**

The paper proposes a resource-efficient method to improve multilingual safety alignment in LLMs. The core contribution is a Multi-Lingual Consistency (MLC) loss, which enforces representational collinearity among semantically equivalent prompts across languages. The method aligns multilingual internal representations toward a shared semantic direction, improving safety consistency without requiring response-level supervision in low-resource languages.

Experiments on Qwen and Gemma models demonstrate substantial improvements in safety rates, especially for low-resource languages, while maintaining general capabilities. The approach is presented as plug-and-play, compatible with existing alignment paradigms such as DPO and SFT, and efficient in data usage.

**Strengths:**

- Clear motivation: Addresses a real and underexplored challenge: multilingual imbalance in LLM safety alignment.

- Conceptual simplicity: The MLC loss is an elegant addition that can integrate easily with existing pipelines.

- Empirical breadth: Includes multiple backbones (Qwen, Gemma), alignment paradigms (DPO, SFT, SimPO, ORPO), and both in- and out-of-distribution tests.

- Data efficiency: Claims strong multilingual gains with minimal additional data (∼1.8M tokens vs. 15M+ for comparable baselines).

- Consistency analyses: Representation-space visualizations (Gram matrices) and PAG metrics provide insightful evidence of improved cross-lingual alignment.

**Weaknesses:**

- Incremental contribution: The MLC loss is essentially a regularization of multilingual representations, conceptually simple and not a fundamentally new paradigm.

- Theoretical shallowness:  Despite heavy mathematical framing (singular value decomposition, spectral view), the theoretical section adds little genuine insight beyond enforcing collinearity.

-  Experimental bias: Evaluations rely on safety datasets constructed in English, potentially conflating multilingual improvement with translation artifacts rather than genuine alignment.

**Questions:**

Please see the weaknesses part.

---

> ### Author Response · Authors · 2025-11-20
> **Response to Reviewer 94Ju**
>
> We sincerely thank the reviewer for the valuable time and their positive assessment of our work. We address the concerns raised in the Weaknesses section as follows.
>
> 1. **Contribution Discussion (Weakness 1)**: We appreciate the reviewer’s observation that MLC is conceptually simple. Indeed, simplicity and modularity are core design goals of our method: we intentionally aim for a loss function that can be cleanly integrated into existing alignment pipelines with minimal engineering overhead. However, we respectfully argue that the contribution is not merely incremental for the following reasons:
>
>     * Solving an Underexplored Challenge: As noted in your Strengths, our work effectively addresses the real and underexplored challenge of multilingual imbalance in LLM safety alignment. The MLC loss provides a resource-efficient and generalizable solution that significantly closes the performance gap in low-resource settings.
>
>     * Algorithmic Innovation in Design: The MLC loss is an elegant addition that achieves complex cross-lingual knowledge transfer through a simple, plug-and-play mechanism. We consider this a substantial algorithmic innovation in the design of multilingual alignment objectives. Its strength lies in its simplicity and its seamless integration into various existing alignment pipelines, demonstrating its wide applicability and practical value.
>
>     We believe that designing an effective, minimal-overhead mechanism to address a critical real-world problem constitutes a significant contribution to the field.
>
>
> 2. **Theory Discussion (Weakness 2)**: Thank you for your sincere comment. Collinearity is straightforward, serving as the ultimate goal for multilingual safety alignment from a representation perspective. However, how to achieve this is non-trivial. Beyond collinearity, our method provides further insights within our progressively theoretical derivation. First, we frame the multilingual alignment problem to the similarity among multilingual representations in queries. This formulation not only provides the foundation for the following theoretical analysis but also motivates future related work in a new and principled way. Second, we provide a new perspective to implement the similarity maximizing in rank-1 approximation. This rank-1 approximation indicates enhancing the shared component among different languages as a proxy for similarity. Without operating any specific language to another, we can directly align all at once. This result is inspiring for achieving collinearity. Following this, we thus propose our learning objective by connecting rank-1 to singular values. This is supported by the Optimal Approximation Theorem. Our theory provides more to approach collinearity rather than simply introducing this. We also add a transition after the proposition to clarify our theoretical derivation.
>     How to ground this into practical learning with principles is the key philosophy of our method design, especially the theoretical derivation. Our theoretical derivation is rigorous with simplicity,  and elegant to explain the underlying mechanism and making it easy for the reader to follow.
>
>
> 3. **Experimental Setting Discussion (Weakness 3)**: We appreciate this important point. Existing multilingual safety benchmarks indeed mostly originate from English datasets. Our experimental design follows standard practice in prior work, where the typical way is to construct multilingual safety evaluation sets by translating English test queries into different languages, either through machine translation[1][2][3] or with human annotators[4].
>
>     To provide robust evidence, our evaluations adopt a mixed setup encompassing both methodologies:
>     * Machine-Translated Setup: We build multilingual PKU-SafeRLHF test sets using GPT-4o translations (following [2]).
>     * Human-Translated Setup: We also include results on the Multijail evaluation set, which leverages human-translated queries.
>
>     This setup allows us to demonstrate the effectiveness of our MLC loss across different levels of evaluation fidelity, thereby providing a more reliable validation of genuine multilingual alignment.
>
>
>     [1] Yong, Zheng-Xin, Cristina Menghini, and Stephen H. Bach. "Low-resource languages jailbreak gpt-4." arXiv preprint arXiv:2310.02446 (2023).
>
>     [2] Zhao et al. "MPO: Multilingual Safety Alignment via Reward Gap Optimization." In Proceedings of the 63rd Annual Meeting of the Association for Computational Linguistics (Volume 1: Long Papers), pages 23564–23587, Vienna, Austria. Association for Computational Linguistics.
>
>     [3] Wang, Wenxuan, et al. "All languages matter: On the multilingual safety of LLMs." Findings of the Association for Computational Linguistics: ACL 2024. 2024.
>
>     [4] Deng, Yue, et al. "Multilingual Jailbreak Challenges in Large Language Models." The Twelfth International Conference on Learning Representations.

---

### Official Review · Reviewer_xbQU · 2025-11-02

**Soundness:** 3
**Presentation:** 3
**Contribution:** 3
**Rating:** 6
**Confidence:** 3

**Summary:**

This paper tackles the challenge of multilingual safety alignment in large language models (ensuring models refuse harmful prompts consistently across languages). The authors propose a Multilingual Consistency (MLC) loss that complements existing post-training methods such as SFT or DPO. The loss encourages shared multilingual representations by promoting collinearity across query embeddings in different languages. Specifically, for each multilingual query set, the hidden representation of the last token in each language is linearly projected, normalized, and stacked into a matrix. The method minimizes the distance of this matrix from its best rank-1 approximation, derived via its top singular value, effectively enforcing a shared semantic direction.

The approach does not require multilingual responses, only crosslingual prompts, making it resource-efficient. Experiments on two safety benchmarks (PKU-SafeRLHF and MultiJail) across ten languages show substantial improvements in multilingual safety consistency, especially for low-resource languages, while maintaining general capabilities on MMLU.

**Strengths:**

- **Important problem:** The paper addresses the problem of ensuring safe and consistent behavior across languages in LLMs.
- **Conceptually elegant and technically sound:** The proposed spectral regularization via rank-1 optimization is simple yet well motivated and theoretically grounded.
- **Strong empirical results:** Comprehensive evaluations across datasets, languages, and base alignment paradigms demonstrate consistent gains, especially for low-resource settings.
- **Practical and efficient:** The method is plug-and-play, adds minimal computational cost, and does not require multilingual response data.

**Weaknesses:**

- **Weak related work discussion:** The discussion of multilingual alignment baselines (e.g., MPO, SDRRL) is both incomplete and difficult to follow. The main paper only names them without explanation, forcing readers to consult the appendix, which is itself hard to follow. As a result, it is difficult to understand how these baselines differ conceptually or why they are appropriate points of comparison.
- **Limited baselines:** The paper lacks an upper-bound comparison, e.g., training with fully translated safety data across languages, to contextualize achievable performance ceilings.
- **Evaluation of general capabilities is narrow:** The use of MMLU alone (amultiple-choice benchmark) provides a limited view of cross-lingual reasoning and generation quality. More generative evaluations could clarify whether safety alignment affects multilingual fluency or reasoning.
- **Missing ablation studies:** The paper would benefit from ablations isolating the contribution of (i) the linear projection, (ii) the choice of singular-value-based regularization versus alternatives such as cosine similarity, and (iii) the temperature parameter τ.

**Questions:**

- **Linear projection:** Is the linear extractor $W$ trained jointly with the model? Please clarify in Section 3.2.
- **Baselines:** Briefly summarize how MPO and SDRRL operate in the main paper, this would make the comparison more self-contained.
- **Ablations:** Could you provide results using alternative similarity measures (e.g., cosine loss) or removing the linear projection to test sensitivity?
- **Capability evaluation:** MMLU is a multiple-choice benchmark and thus does not assess generation abilities or language control. I suggest adding a CoT-style evaluation where the model must generate reasoning in the target language. This would allow you to measure both accuracy and linguistic consistency (e.g., avoiding language mixing), providing a more complete picture of whether the proposed training preserves generative behavior across languages.

---

> ### Author Response · Authors · 2025-11-20
> **Response to Reviewer xbQU (Part I)**
>
> We sincerely thank the reviewer for the valuable time and constructive suggestions, which have helped us improve the clarity and completeness of our paper. Below we provide point-by-point responses and clarify the improvements made in the revised version.
>
> 1. **Related Work (Weakness 1 & Question 2)**: Thank you for your kind reminder. We have revised Section 4.1 in the main paper to briefly summarize MPO and SDRRL and clarify the conceptual distinction and rationale for choosing them, thusing enhancing the undertandablility:
> *SDRRL* as the representative data-augmentation-based multilingual alignment method.
> *MPO* as the representative algorithm-design-based multilingual safety alignment approach.
> Moreover, we also provide clearer procedural steps and implementation details of both MPO and SDRRL in Appendix D.3.
>
> 2. **Upper-Bound Comparison (Weakness 2)**: Your suggestion is quite insightful! To better contextualize the performance of our MLC method, we agree that including a strong upper-bound baseline is important.
> Following your advice, we add a new baseline that intuitively approximates the expected performance ceiling for each language:
> DPO trained on fully translated safety data across all target languages.
> The results are as follows:
>
>     | Model                                |        en |        zh |        ru |        ja |        ar |        bn |        sw |        ur |        ps |        ku |     Avg Safety Rate ↑ |
>     | ------------------------------------ | --------: | --------: | --------: | --------: | --------: | --------: | --------: | --------: | --------: | --------: | --------: |
>     | Raw                                  |     93.33 |     96.11 |     93.33 |     92.22 |     93.89 |     53.33 |      6.11 |     33.89 |     21.11 |     12.22 |     59.55 |
>     |Full Translation | 99.44 |   98.89 |     98.33 |      98.33|        100.00 |    89.44 |    20.00 |    73.33 |   65.56 |   41.67 |         78.50 |
>     | MLC (ours)  | 99.44 | 96.67 | 97.78 | 98.33 | 98.33 | 95.00 | 92.78 | 92.78 | 91.11 | 97.22 | 95.94 |
>
>     We observe that our MLC method can achieve a performance level comparable to, or even higher (on specific language) than, the Full-Translated baseline. It suggests that the translation-based distillation may have intrinsic information loss, especially for low-resource languages.

---

> > ### Author Response · Authors · 2025-11-20
> > **Response to Reviewer xbQU (Part II)**
> >
> > 3. **Evaluation of General Capabilities (Weakness 3 & Question 4)**: Thank you for pointing out the importance of evaluating general capabilities. To provide a more complete picture of whether our training preserves cross-lingual reasoning and generative behavior, we have supplemented the evaluation with MGSM (for CoT-style reasoning) and MT-Bench (for open-ended generation and conversational quality).
> >
> >     *MGSM*: A multilingual benchmark of grade-school math problems that requires models to leverage step-by-step CoT reasoning. It contains 250 GSM8K problems, each translated into 10 languages by human annotators.
> >
> >     *MT-Bench*: Measures the ability of large language models (LLMs) to engage in coherent, informative, and engaging conversations.The benchmark consists of 80 high-quality conversational prompts and employs the LLM-as-a-Judge. In our experiments, GPT-4o is prompted to assign a score from 1 to 10 to each model response. We use publicly available data in English, Japanese, Chinese, Russian, and Arabic from HuggingFace and GitHub. For low-resource languages (bn, ps, sw, ku, ur), we follow [1] and use GPT-4o to translate the original English MT-Bench queries.
> >
> >     Multilingual General Capability Evaluation (Qwen2.5-7B)
> >     | **Eval**       | **Raw** | **SDRRL** |   **MPO** |   **DPO** | **DPO+MLC** |
> >     | -------------- | ------: | ----: | ----: | ----: | ----------: |
> >     | **MMLU**       |   76.37 | 31.57 | 75.90 | 76.22 |   76.30 |
> >     | **MMMLU-lite** |   55.16 | 49.88 | 52.31 | 54.59 |   52.66 |
> >     | **MGSM**       |   69.89 | 19.31 | 69.09 | 70.61 |   72.84 |
> >     | **MT-Bench**   |   5.965 | 4.738 | 5.806 | 6.025 |   6.726 |
> >
> >     Multilingual General Capability Evaluation (Gemma-2-9B-it)
> >     | **Eval**       | **Raw** | **SDRRL** |   **MPO** |   **DPO** | **DPO+MLC** |
> >     | -------------- | ------: | ----: | ----: | ----: | ----------: |
> >     | **MMLU**       |   74.34 | 51.90 | 73.85 | 73.79 |   73.58 |
> >     | **MMMLU-lite** |   49.25 | 49.65 | 51.06 | 49.27 |   54.61 |
> >     | **MGSM**       |   80.55 | 66.47 | 79.75 | 80.18 |   80.44 |
> >     | **MT-Bench**   |    7.01 | 6.135 | 7.143 | 7.022 |   7.219 |
> >
> >     As expected, our method maintains or even improves performance on complex reasoning (MGSM) and generative tasks (MT-Bench), demonstrating that the proposed consistency loss is effective without sacrificing the model's general multilingual fluency or reasoning capabilities.
> >
> >
> >     [1] Zhao et al. "MPO: Multilingual Safety Alignment via Reward Gap Optimization." In Proceedings of the 63rd Annual Meeting of the Association for Computational Linguistics (Volume 1: Long Papers), pages 23564–23587, Vienna, Austria. Association for Computational Linguistics.
> >
> > 4. **Ablation Study on Linear Projection and Training (Weakness 4-i & Question 1, 3)**: We clarify in the revised Section 3.2 that the linear extractor $W$ is trained jointly with the LLM during the fine-tuning phase.
> >
> >     We also perform an ablation study to isolate the contribution of the linear projection by comparing against using the raw hidden states (non projection) as the reviewer suggested. The results are shown below (here we present safety performance Avg, PAG and general capability metrics only for key information presentation):
> >
> >     | Model                        |     Avg Safety Rate ↑ |   PAG ↑ | MMLU ↑ | MMMLU-lite ↑ |
> >     | ---------------------------- | --------: | ---------: | -----: | -----------: |
> >     | Raw                          |  59.55 |   0.5037 |  76.37 |        55.16 |
> >     | No projection            |98.11  | 0.9733|  76.55 |    47.35 |
> >     | Linear projection (ours) | 95.94 |     0.9697 |  76.30 |        52.66 |
> >
> >     The non projection approach yields a marginally higher safety score ($98.109$), but this significantly compromises multilingual general capabilities ($47.35 \rightarrow 52.66$ on MMMLU-lite). This demonstrates that: Directly imposing consistency on the raw hidden states creates a "harder" constraint, boosting safety but causing more severe interference with the model's general representation.
> >
> >     Furthermore, we discuss the performance of other alternative extractors in the Appendix E.1.

---

> > > ### Author Response · Authors · 2025-11-20
> > > **Response to Reviewer xbQU (Part III)**
> > >
> > > 5. **Ablation Study on Similarity Measures (Weakness 4-ii & Question 3)**:
> > > To demonstrate the choice of singular-value-based regularization, we compare our Rank-1 approximation approach against a standard Cosine Similarity Loss.
> > >
> > >     | Method                |      Avg Safety Rate ↑ |  PAG ↑ | MMLU ↑ | MMMLU-lite ↑ |
> > >     | --------------------- |  ----: | -----: | -----: | -----------: |
> > >     | Raw                   |   59.55 | 0.5037 |  76.37 |  55.16 |
> > >     | Cosine similarity | 98.05  | 0.9714 |  76.17 |    45.92 |
> > >     | Rank-1 (ours)     |  95.94 |0.9697 |  76.30 |        52.66 |
> > >
> > >     Actually, the core idea of our method is to maximize the similarity of multilingual representations. The Cosine Similarity Loss, being a direct measure of pairwise similarity, constitutes a "harder" form of consistency constraint compared to our Rank-1 approximation loss.
> > >     Similar to the projection ablation, the harder constraint of Cosine Similarity achieves slightly higher safety ($\mathbf{98.054}$ vs $95.940$) but again leads to a significant drop in general capability ($\mathbf{45.92}$ vs $\mathbf{52.66}$ on MMMLU-lite). This further validates our design choice, showing that the Rank-1 approximation is a better-balanced loss that achieves high safety with minimal utility degradation.
> > >
> > >
> > > 6. **Hyperparameter Analysis of Temperature Parameter $\tau$ (Weakness 4-iii)**:
> > > We have added an analysis of the temperature parameter $\tau$ to demonstrate its impact on the strength of the consistency signal. Theoretically, a smaller $\tau$ leads to a sharper softmax, resulting in a less constrained (smaller) loss, while a larger $\tau$ results in a smoother softmax, leading to a stronger constraint signal. Our experimental results validate this:
> > >
> > >     | $\tau$              |       Avg Safety Rate ↑ |        PAG ↑ | MMLU ↑ | MMMLU-lite ↑ |
> > >     | -------------- |  --------: | ---------: | -----: | -----------: |
> > >     | 0.01           |    65.28 |          0.54 |  76.27 |        54.89 |
> > >     | 0.1            |    91.17 |      0.8517 |  75.97 |        57.56 |
> > >     | 0.2(default)  |  95.94 |       0.9697 |  76.30 |        52.66 |
> > >     | 0.5            |      98.28 | 0.9838 |  76.52 |    39.11 |

---

### Author Response · Authors · 2025-12-01
**General Comments from Authors**

Dear PC, SAC, AC, and Reviewers,

We sincerely thank the reviewers for their valuable time and constructive suggestions.
Our manuscript has received four initial reviews, scored as 6664, where only one borderline negative review (4TKF) was provided.
We are greatly encouraged by the reviewers' positive recognition of our work, and motivated to address reviewers' concerns. Most reviews recognize the novelty, theoretical soundness, empirical effectiveness, and practical efficiency of our method. Specifically,
Reviewers xbQU and 94Ju noted the **clear motivation** and acknowledged **the importance of the problem** (*i.e.,* multilingual imbalance in LLM safety alignment). Reviewers xbQU, 94Ju, and 4TKF highlighted the **conceptual simplicity** and **theoretical soundness** of our spectral regularization via rank-1 optimization, noting that it is *clean, well-motivated, and easy to integrate into existing pipelines*. Reviewers xbQU, 94Ju, and 4nM1 further commended the **strong and comprehensive empirical results** across datasets, languages, base models, and alignment paradigms, particularly the consistent improvements in low-resource languages and preservation of performance in high-resource settings. All reviewers (xbQU, 94Ju, 4TKF, 4nM1) remarked on the **practical efficiency** of our method, including its plug-and-play usage and reliance only on translated prompts rather than response-level labels.

At the same time, reviewers shared similar questions primarily related to clarifications and auxiliary analyses.
In particular, reviewer 4TKF with borderline negative score mainly requested further clarifications on details of representation extraction, loss objective derivation, and hyperparameter choices. Most details are already included in Appendix and we have re-organized them to the main text for better accessibility.

We have carefully addressed all reviewer questions in detailed, point-by-point responses, and have incorporated their suggestions into the revised manuscript. The major improvements (highlighted in blue in the revised PDF) include:

* **Enhanced Clarity and Implementation Details**:
We clarified the training procedure of the projection module, explained the choices of our baselines, and improved the conceptual flow of the theoretical motivation and practical implementation.

* **Content Re-organization**:
Given multiple reviewers' interest in the effect of layer depth, we moved the Layer Analysis (previously Appendix E.1) into the main text as Section 4.7, making the insights about layer selection more accessible and easier to follow.

* **Additional analyses and ablations**：
In response to reviewer suggestions, we added several new analyses in the Appendix, such as ablations on similarity measures, the linear projection design, and the influence of the temperature parameter $\tau$.

We hope that the revised manuscript, complete with these clarifications and additional empirical evidence, addresses all the reviewers' concerns. It is a pity that, due to the OpenReview API Security Incident on November 27, the reviewers were unable to provide further feedback. We hope the above general comments can help the AC gain a clearer and more direct understanding of the contributions and revisions of our work.

Sincerely,

The Authors

---

### Meta-Review · Area_Chair_Jdb5 · 2026-01-01

**Summary:**

This paper proposed a regularizer to solve multilingual alignment problems. There were various concerns, including limited baselines, limited evaluation of general capabilities, missing ablations, the layer selection is a bit unclear and without too much analysis. There were also concern over the issue of using English-derived dataset in multilingual alignment studies.

**Reviewer Concerns:**

The authors provided additional upper baseline results, more evaluations on general capabilities and additional ablations, hyper parameter studies, etc. The author also provided clarification to many questions on the presentation. Those addressed many of the concerns raised by the reviewers. The concern of using English-derived dataset in multilingual alignment studies is not fully addressed, but this has been an issue of many previous paper in the literature as well and the value of this paper outweigh this weakness.

**Reviewer Scores:**

I expect most of the reviewers to keep their positive score or raise slightly. In particular, reviewer 4TKF who was giving weak rejection may increase their score given the clarification from the authors to their questions.

---

### Decision · Program_Chairs · 2026-01-26

Accept (Poster)